# Kinetics of Catalytic Oxidation of Methylene Blue with La/Cu Co-Doped in Attapulgite

**DOI:** 10.3390/ma16052087

**Published:** 2023-03-03

**Authors:** Jianping Shang, Wei Zhang, Zhengliang Dong, Hua-Jun Shawn Fan

**Affiliations:** 1College of Chemical Engineering, Sichuan University of Science and Engineering, Zigong 643099, China; 2Sichuan Institute of Geological Survey Salt Geology Drilling Team, Chengdu 610059, China

**Keywords:** attapulgite, methylene blue, catalytic, kinetics

## Abstract

Methylene blue (MB) is a common pollutant in wastewater from the printing and dyeing industries. In this study, attapulgite (ATP) was modified with La^3+^/Cu^2+^, using the method of equivolumetric impregnation. The La^3+^/Cu^2+^ -ATP nanocomposites were characterized using X-ray diffraction (XRD) and scanning electron microscopy (SEM). The catalytic properties of the modified ATP and the original ATP were compared. At the same time, the influence of the reaction temperature, concentration of methylene blue and pH on the reaction rate were investigated. The optimal reaction conditions are as follows: MB concentration is 80 mg/L, the dosage of the catalyst is 0.30 g, the dosage of hydrogen peroxide is 2 mL, the pH is 10 and the reaction temperature is 50 °C. Under these conditions, the degradation rate of MB can reach 98%. The recatalysis experiment was carried out reusing the catalyst, and the experimental results showed that the degradation rate could reach 65% after three uses, indicating that the catalyst could be recycled many times and costs could be reduced. Finally, the degradation mechanism of MB was speculated, and the reaction kinetic equation was obtained as follows: −dc/dt = 14,044 exp(−3598.34/T)C(O)^0.28^.

## 1. Introduction

Lanthanum is one of the more common rare earth elements. It is easily oxidized to a white oxide powder in the air [1]. Supported lanthanum oxide and lanthanum-based mixed oxides are effective catalysts for the removal of organics from wastewater [2,3].

Nowadays, printing and dyeing wastewater is characterized by high alkalinity, complex composition and difficulty in treatment [4]. At present, the treatment methods of printing and dyeing wastewater are as follows: adsorption [5,6,7], catalytic oxidation [8,9,10,11,12] and biological decomposition [13,14]. The catalytic wet oxidation (CWO) process is usually carried out under a high temperature and pressure, which limits its application. Therefore, the development of a new process and the preparation of new catalysts with high efficiency are the main directions for improving catalytic oxidation efficiency.

Attapulgite (or palygorskite, as it is commonly known) is a crystalline magnesium aluminosilicate hydrate, formulated as Si_8_O_20_(Mg, Al, Fe)_5_(OH)_2_(OH_2_)_4_·4H_2_O. ATP is a natural nanomaterial with active hydroxyl groups on the surface and zeolite-like channels inside [15]. It has a large specific surface area and is often used as an adsorbent, catalyst and catalyst carrier [16,17,18,19]. ATP is suitable as a catalyst carrier because of its large specific surface area, high chemical stability and mechanical stability. Especially for its low price, ATP has practical significance in wastewater treatment. Zhang et al. [20] prepared lanthanum-modified mesoporous adsorption materials using the hydrothermal synthesis method and found that the modified materials greatly improved the adsorption capacity of arsenic and phosphorus in an adsorption experiment. Hamed et al. [21] prepared lanthanum-doped CuCr_2_O_4_ nanoparticles using the sol–gel method. The photocatalytic experiments on organic pollutants showed that the degradation rate of nanomaterials mixed with lanthanum reached 98%, which was 20% higher than that of unmixed catalysts. Wang et al. [22] added lanthanum to a V_2_O_5_ catalyst to investigate the effect of lanthanum on its catalytic performance. The experimental results showed that the catalyst can participate in the reaction at a low temperature and that a certain amount of lanthanum can promote the formation of pyrosulfate and increase the specific surface area and pore diameter of the catalyst. In this paper, attapulgite (ATP) was modified using La^3+^/Cu^2+^, through the method of equivolumetric impregnation. The effect of the reaction temperature, catalyst dosage, pH, and hydrogen peroxide dosage on the degradation rate of MB was investigated.

## 2. Materials and Methods

### 2.1. Materials

Attapulgite (ATP) came from Xuyi, Jiangsu. Copper nitrate (Cu(NO_3_)_2_•3H_2_O) was purchased from the Magnolia town industrial development zone, Xindu District, Chengdu. Lanthanum nitrate (La(NO_3_)_3_•6H_2_O) was purchased from Tianjin Kemeo chemical reagent Co., Ltd., Tianjin, China. Hydrogen peroxide (H_2_O_2_, 30%) was purchased from Chengdu cologne chemical Co., Ltd. Methylene blue (C_16_H_18_ ClN_3_S•3H_2_O) was purchased from Chengdu cologne chemical Co., Ltd. These drugs were used for the analysis.

### 2.2. Methods

The experimental set-up and reagents used in this work were described in detail in the literature [23]. The experimental procedure was as follows: about 40 g of attapulgite with a particle size of about 180 mesh was screened and immersed in 200 mL of 4 mol/L dilute hydrochloric acid for 3 h, then the vacuum filter was used to filter, followed by repeated rinse with deionized water until the filtrate was neutral, and the remainder was put in the drying oven, drying under 80 °C for 2 h. Then the composite catalyst was prepared using the equal volume method, in which the mass ratio (ATP:La) was 20:1 and the molar ratio (La:Cu) was 3:1. The loading time was 12 h. The loaded catalyst was put in the drying oven, drying under 80 °C for 6 h. Products were dried in the tube furnace at 300 °C, 400 °C and 500 °C calcine, respectively, for 2 h. The calcined catalyst was sealed and bagged for later use. After the temperature reached the reaction temperature, the La^3+^/Cu^2+^-ATP catalyst and hydrogen peroxide were added, and samples were taken at the times of 0, 1, 3, 5, 7, 10, 15 and 20 min.

Methods for the analysis of MB concentration include colorimetry, iodometry, spectrophotometry and liquid chromatography. For the advantages of simpler operation, lower cost and higher accuracy, spectrophotometry was selected as the MB concentration analysis method in this experiment. By measuring the absorbance or luminous intensity of the substance at a specific wavelength or within a certain wavelength range, a qualitative and quantitative analysis method was adopted for the substance. The Lambert–Beer theorem was applied, in the case of the low concentration solutions.

## 3. Results and Discussion

### 3.1. Effect of Hydrogen Peroxide

For the 250 mL MB solution, the experimental conditions are as follows: the concentration of MB is 80 mg/L, the dosage of the catalyst is 0.3 g, the reaction temperature is 50 °C, the pH is 10, the dosage of hydrogen peroxide is 0 mL, 0.5 mL, 1.0 mL, 1.5 mL and 2.0 mL, respectively. The experimental results are shown in Figure 1.

As can be seen from Figure 1, when there was no H_2_O_2_, MB was adsorbed on the ATP surface, resulting in a decrease in MB and an increase in the degradation rate. When H_2_O_2_ was added, the degradation rate of MB significantly increased, which was due to the effective collision between H_2_O_2_ and the catalyst, the Fenton-like reaction and the formation of •OH, which promoted the decomposition of MB. In addition, with the increase in H_2_O_2_ dosage, the degradation rate of MB accelerated, and the final degradation rate also increased within the same timeframe. Therefore, the dosage of H_2_O_2_ in subsequent experiments was 2 mL.

### 3.2. Effect of Reaction Temperature

For the 250 mL MB solution, the experimental conditions are as follows: the concentration of MB is 80 mg/L, the dosage of the catalyst is 0.3 g, the pH is 10, the dosage of hydrogen peroxide is 2.0 mL and the reaction temperature is 30 °C, 35 °C, 40 °C, 45 °C and 50 °C, respectively. The experimental results are shown in Figure 2.

It can be seen from Figure 2 that, in the case of enough hydrogen peroxide, from 30 °C to 50 °C, with the increase in reaction temperature, the degradation rate of MB rises. This may be because as the temperature increases, the activation ratio in the molecules of hydrogen peroxide increases, and the movement will increase the probability of effective collision, produce more •OH and accelerate the decomposition of methylene blue. In the 30 °C to 50 °C temperature range, the degradation rate of MB increases with the rise of temperature. Therefore, in subsequent experiments, the choice reaction temperature is 50 °C.

### 3.3. Effect of Dosage of Catalyst

For the 250 mL MB solution, the experimental conditions are as follows: the concentration of MB is 80 mg/L, the reaction temperature is 50 °C, the pH is 10, the dosage of hydrogen peroxide is 2.0 mL and the dosage of the catalyst is 0.10 g, 0.15 g, 0.20 g, 0.25 g and 0.30 g, respectively. The experimental results are shown in Figure 3b. Meanwhile, the original soil was compared with the modified ATP. The experimental results are shown in Figure 3a.

As can be seen from Figure 3a, the degradation rate of MB is very low without a catalyst, because hydrogen peroxide hardly reacts with MB. After adding 0.3 g attapulgite, the degradation of MB becomes evident, because ATP is a kind of water-containing crystallization of magnesium aluminum silicate and itself contains certain metal ions. After adding the hydrogen peroxide solution, the Fenton reaction happens, leading to the catalytic decomposition of methylene blue. It has internal holes that can produce adsorption, which has a certain impact. Compared with the original attapulgite, the catalytic degradation effect using the modified ATP is further enhanced because the modified ATP can introduce specific metal ions, which can accelerate the generation of hydroxyl radicals. In the same timeframe, the degradation rate of MB will increase.

It can be seen from Figure 3b that the degradation rate of MB increases with the increase in the amount of modified ATP. Moreover, the higher the amount of catalyst, the faster the initial degradation rate and the higher the final degradation rate. This is due to the fact that, with a certain concentration of hydrogen peroxide and MB, the amount of catalyst increases, and the catalyst can provide more active sites, so that the effective collisions per unit volume increase, the generation rate of •OH increases and the degradation of MB is promoted. Therefore, in subsequent experiments, the dosage of catalyst was determined to be 0.30 g.

### 3.4. Effect of pH

For the 250 mL MB solution, the experimental conditions are as follows: the concentration of MB is 80 mg/L, the dosage of the catalyst is 0.3 g, the reaction temperature is 50 °C, the dosage of hydrogen peroxide is 2.0 mL, and the pH is 4, 7, 8, 9 and 10, respectively. The experimental results are shown in Figure 4.

It can be seen from Figure 4 that the degradation rate of MB under acidic conditions is lower than that under neutral conditions; the reason is that the degradation rate of MB under acidic conditions is inhibited. Because the MB solution exists in the form of cations in water, this weakens the MB under acidic conditions and weakens the electrostatic attraction to the modified catalyst. In alkaline environments, MB increases with the electrostatic attraction to the modified catalyst [24]. This makes the MB respond more rapidly to the modification of the catalyst and, at the same time, MB generates and reacts to the •OH, which make H_2_O_2_ more quickly, prompting the metal on the surface of the modified catalyst to release •OH. Therefore, in a follow-up experiment, the pH is 10.

### 3.5. Effect of Concentration of MB

For the 250 mL MB solution, the experimental condition are as follows: the dosage of the catalyst is 0.3 g, the reaction temperature is 50 °C, the dosage of hydrogen peroxide is 2.0 mL, the pH is 10 and the concentration of MB is 60 mg/L, 70 mg/L, 80 mg/L, 90 mg/L and 100 mg/L, respectively. The experimental results are shown in Figure 5.

As can be seen from Figure 5, with the increase in MB concentration, the trend of the increasing MB degradation rate becomes slower. The higher the concentration of MB, the smaller the final degradation rate. This is because the amount of H_2_O_2_ is the same; thus, in the unit volume solution, the amount of •OH produced is constant, which means that the amount of MB that can catalyze degradation is constant. The lower the concentration, the smaller the amount of MB in the unit volume solution and the more complete the degradation. The higher the concentration, the more MB per unit volume, and the less complete the degradation. Moreover, H_2_O_2_ will decompose itself, which means that H_2_O_2_ is not completely used for the catalytic degradation of MB. Therefore, in subsequent experiments, the concentration of MB was 80 mg/L.

### 3.6. Effect of Calcination Temperature

According to the above experiments, the best experiment conditions are as follows: the MB concentration is 80 mg/L, 250 mL, the dosage of the catalyst is 0.30 g, the dosage of hydrogen peroxide is 2 mL, the pH is 10 and the reaction temperature is 50 °C. Under these experiment conditions, the effect of calcination temperature on the catalyst was investigated. The experimental results are shown in Figure 6.

As you can see from Figure 6, a calcination temperature of 500 °C for the modified catalyst is better than that of 300 °C and 400 °C. The calcination temperatures of 300 °C and 400 °C for the modified catalyst have a similar effect. This is because in the 500 °C calcination, the crystallization water almost completely removes the ATP [25], expanding the internal space of the ATP and further increasing the specific surface area. H_2_O_2_ can interact more quickly and easily with modified metal on the surface of the catalyst when effective collision occurs, so that OH is produced more quickly, accelerating the degradation of MB. Where the calcination temperature is 300 °C or 400 °C, the modified catalyst is removed from the channel of free water by a small amount of crystal water; thus, the ATP interior hole does not get bigger.

### 3.7. Regeneration Experiment

Under the best experiment conditions, the experiment to reuse the catalyst was conducted.

As can be seen from Figure 7, the catalytic decomposition rate of MB was relatively high in the first place. With the increase in calcination temperature, the degradation rate of MB was 95%, 96% and 97%, respectively. The degradation rate of MB using the reused catalyst is also relatively good. With the increase in calcination temperature, the degradation rate can reach 76%, 78% and 80%, respectively, indicating that the catalyst has good secondary catalytic performance and can be recycled. After the third use, the degradation rate of MB is still around 65%. The results showed that the reuse ability of the catalyst was good.

### 3.8. Characterization

The ATP and modified ATP were characterized using SEM, with a voltage of 15 KV and a magnification of 2000 and 5000 times, respectively. The obtained images were as follows.

As can be seen from Figure 8a, the original soil is larger than the modified ATP at 2000 times magnification, with many folds and protrusions on the surface. It can be seen from Figure 8b that the size of the La^3+^/Cu^2+^ -ATP is around 30 μm. Compared with the original ATP, the surface of the modified ATP is relatively smooth, which may damage the outer structure and cause some surface structures to fall off during the acid soaking and calcining stages. As can be seen from Figure 8c, the original ATP soil has internal depressions, but its diameters are all large. As can be seen from Figure 8d, the surface of the modified ATP has a lump, which is presumed to be a metal material loaded on it. In addition, the diameter of the surface of the modified ATP becomes larger towards the inner depression, which may be due to the removal of water from the ATP during calcining.

The ATP and modified ATP were characterized using XRD, and the changes of materials in ATP and modified ATP were compared. The XRD results are shown in Figure 9.

It can be seen from Figure 9 that a strong diffraction characteristic peak of lanthanum appears when 2θ is at 34.3° and a diffraction characteristic peak of copper appears when 2θ is at 39.4°. Other peaks in the graph are caused by the presence of impurities in the sample itself. The overall crystal lattice of the catalyst is intact and not destroyed. The product has obvious diffraction peaks and the interlayer spacing fluctuates.

### 3.9. The Reaction Mechanism

Oxidation of the MB solution by the ATP rod–soil occurs through a heterogeneous catalytic oxidation mechanism. With hydrogen peroxide in the solution, the reaction of MB is divided into two kinds. One is the direct reaction of hydrogen peroxide and MB through direct action; the results are shown in Figure 10 (experimental conditions: reaction temperature: 40 °C, pH: 7, H_2_O_2_: 2 mL, MB: 50 mg/L, 250 mL and the modification of the catalyst: 0 g).

The degradation rate of MB is less than 5%. This almost indicates that H_2_O_2_ does not react directly with MB. The other kind is the chain reaction of H_2_O_2_ in the solution to form the highly oxidizing •OH, which catalyzes the decomposition of MB. At the same time, CuO is contained in the catalyst and part of the CuO will enter the lattice of La_2_O_3_ [26]. A small number of lanthanum ions will be replaced by copper ions, resulting in more empty oxygen positions [27,28]. The presence of empty oxygen potential will participate in the decomposition of H_2_O_2_. According to the mechanism proposed by Deraz [29], the reaction pathway of H_2_O_2_ decomposition in the La–Cu mixed oxide system can be proposed, and its surface is exposed to the La^2+^/La^3+^, Cu^2+^/La^2+^, Cu^+^/Cu^2+^ ion pairs:H_2_O^2^ ↔ H^+^ + HO_2_^−^(1)
Cu^+^ → Cu^2+^ + e(2)
H^+^+e → H(3)
HO_2_^−^ → HO_2_ + e(4)
La^3+^ + e → La^2+^(5)
Cu^+^ + La^3+^ → Cu^2+^+La^2+^(6)
H_2_O_2_ ↔ H++HO_2_(7)
La^2+^ → La^3+^ + e(8)
H^+^ + e → H(9)
HO_2_^−^ → HO_2_ +e(10)
Cu^2+^ + e → Cu^+^(11)
Cu^2+^ + La^2+^ → Cu^+^ + La^3+^(12)

Based on the above information, it is speculated that the degradation of MB is as follows:La^2+^ + H_2_O_2_ → La^3+^ + •OH + OH^-^(13)
Cu^+^ + H_2_O_2_ → Cu^2+^ + •OH + OH^−^(14)
•OH + H_2_O_2_ → HO_2_• + H_2_O(15)
La^3+^ + HO_2_• → La^2 +^ + HO_2_^−^(16)
Cu^2+^ + HO_2_• → Cu^+^ + HO_2_^−^(17)
Cu^+^ + La^3+^ → Cu^2+^ + La^2+^(18)
MB + •OH → CO_2_ + H_2_O(19)

As the main active substance, La^2+^ firstly initiates the REDOX reaction to produce •OH (Equation (13)). It is reported that Cu+ also has Fenton-like properties and can catalyze the activation of H_2_O_2_ to generate •OH (Equation (14)). At the same time, Cu^2+^ can react with the HO_2_• generated in the reaction to form Cu^+^, achieving the purpose of the Cu^2+^ and Cu+ cycle (Equation (17)). In addition, as previously mentioned, La^3+^ was reduced by Cu^+^ to La^2+^ (Equation (18)). Thus, the reaction, to some extent, increased the content of La^2+^ in the Fenton reaction, promoting the generation of •OH in the reaction process, which is likely to be a key step in improving the catalytic activity of the catalyst.

### 3.10. Reaction Kinetic

The catalytic oxidation of hydrogen peroxide has many applications in the oxidation and decomposition of organic impurities in wastewater. In the catalytic oxidation system, temperature, time, catalyst, substrate and the amount of oxidant are the main factors affecting the oxidation rate.

The reaction rate equation is as follows:(20)$$-\mathrm{dC}/\mathrm{dt}=k(C_{\mathrm{MB}}{)}^{m}$$

Here, k represents the reaction rate constant, C represents the concentration of methylene blue in solution, mol/L and m represents the reaction series.

The reaction conditions are as follows: the temperature is 50 °C, the concentration of MB is 80 mg/L and the reaction rate is simulated using MATLAB software. According to the results of the fitting theory, the dynamics equation is as follows:(21)$$-\mathrm{dC}/\mathrm{dt}=6.027\;\times \;(C{)}^{2.45}$$

According to the experimental data on the effect of reaction temperature on MB degradation mentioned above, MATLAB software was used for data fitting, and the results are shown in Table 1.

According to Arrhenius’ formula:(22)$$k=k0_{\;}\exp(-\mathrm{Ea}/\mathrm{RT})$$
where k_0_ represents the preexponential factor and dimension is related to the reaction series m; Ea represents the reaction activation energy, J/mol; R represents the ideal gas constant, 8.314 J/(mol•K); and T represents the reaction temperature, K. The results are shown in Figure 11.

The intercept is 11.82. That is, lnk_0_ = 11.82, k_0_ = e^11.82^ = 1.36 × 10^5^ and the slope is −3241.32, (−Ea/R) = −3241.32, Ea = 26.95 kJ/mol. Therefore, k = 1.36 × 10^5^ exp (3241.32/T).

The kinetic equation of methylene blue catalytic oxidation is as follows:−dC/dt = 1.36 × 10^5^ exp(3241.32/T)(C_MB_)^2.45^(23)

The final degradation rate can reach 98% under optimum conditions. This result indicates that the above kinetic equation is suitable for most reactions. Meanwhile, the degradation rate of MB in this process is greater than the 73.8% rate obtained by other researchers under the optimum conditions [30].

## 4. Conclusions

Through preparing the modified ATP, the influencing factors of MB degradation were investigated, examining various aspects. ATP is an excellent catalyst carrier with capacities for adsorption and catalytic oxidation. Compared with ATP, the modified ATP improved its catalytic performance. The optimum experimental conditions obtained from the experiment are as follows: reaction temperature: 50 °C, pH: 10, hydrogen peroxide dosage: 2 mL and modified catalyst: 0.3 g. This technology will provide certain theoretical support for the industry.

## Figures and Tables

**Figure 1 materials-16-02087-f001:**
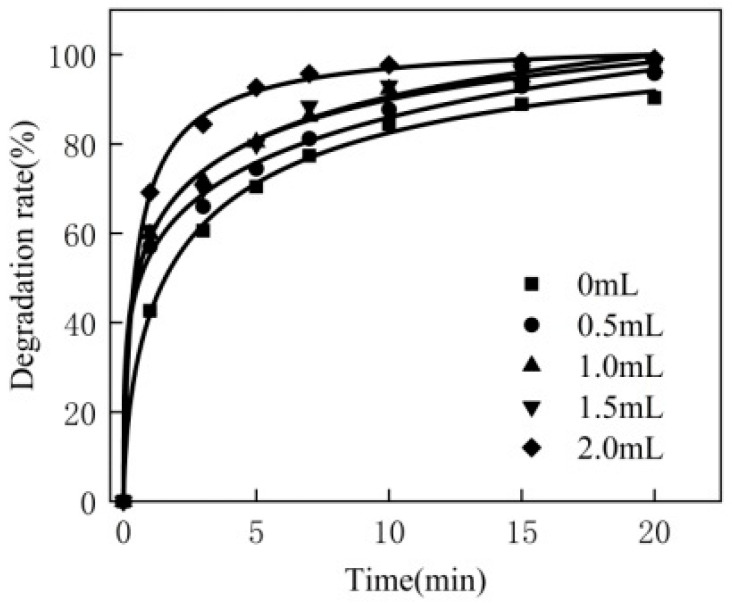
Effect of hydrogen peroxide consumption.

**Figure 2 materials-16-02087-f002:**
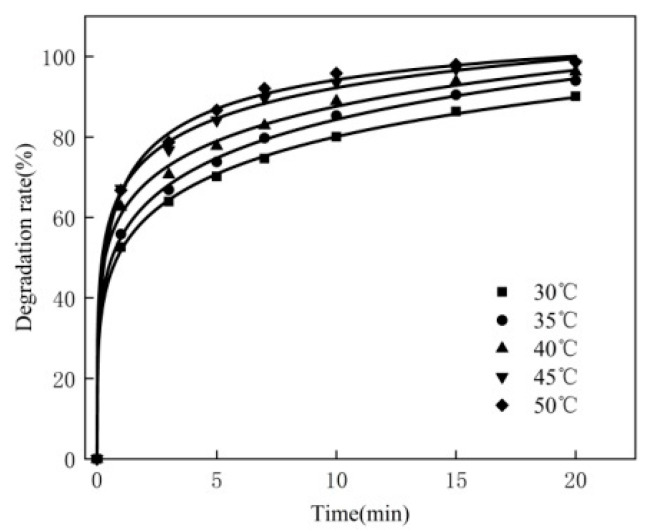
Effect of reaction temperature.

**Figure 3 materials-16-02087-f003:**
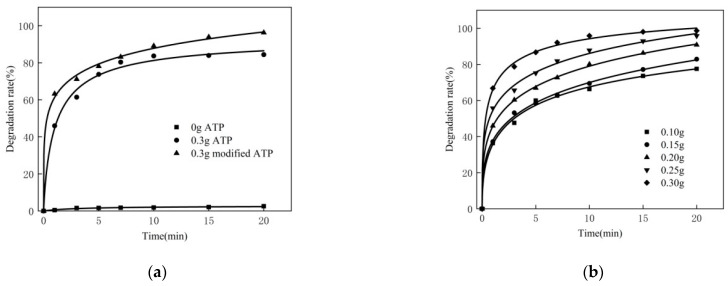
(**a**) Performance of modified ATP and ATP. (**b**) Effect of dosage of catalyst.

**Figure 4 materials-16-02087-f004:**
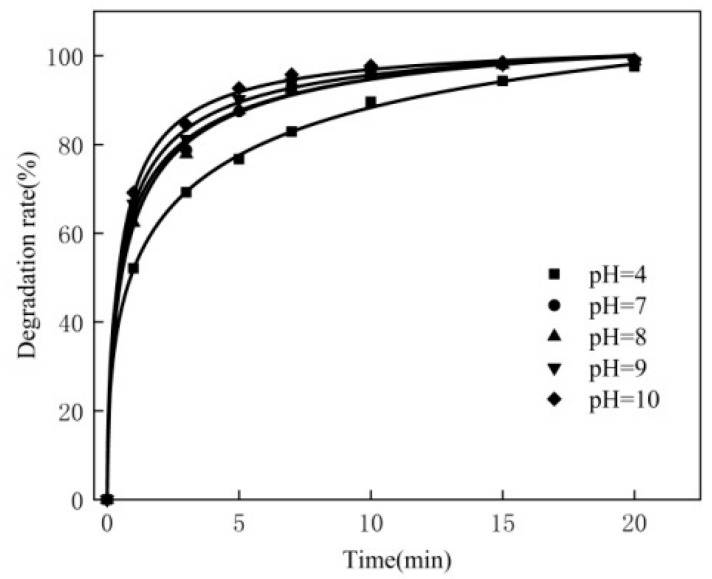
Effect of pH.

**Figure 5 materials-16-02087-f005:**
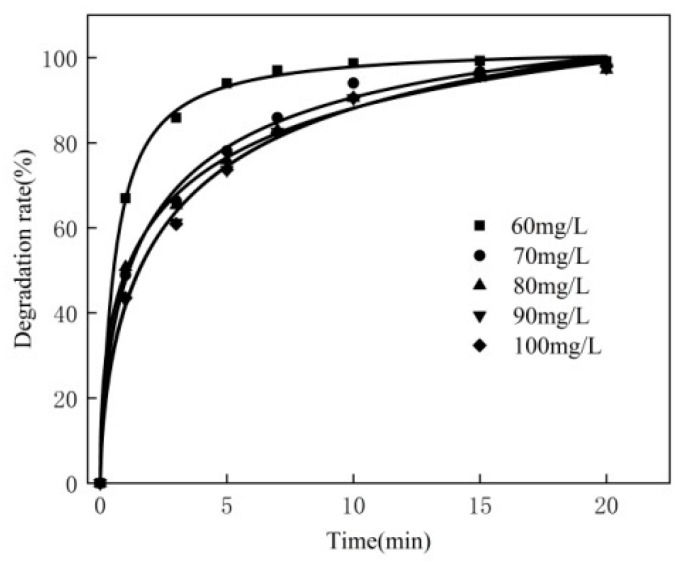
Effect of MB concentration.

**Figure 6 materials-16-02087-f006:**
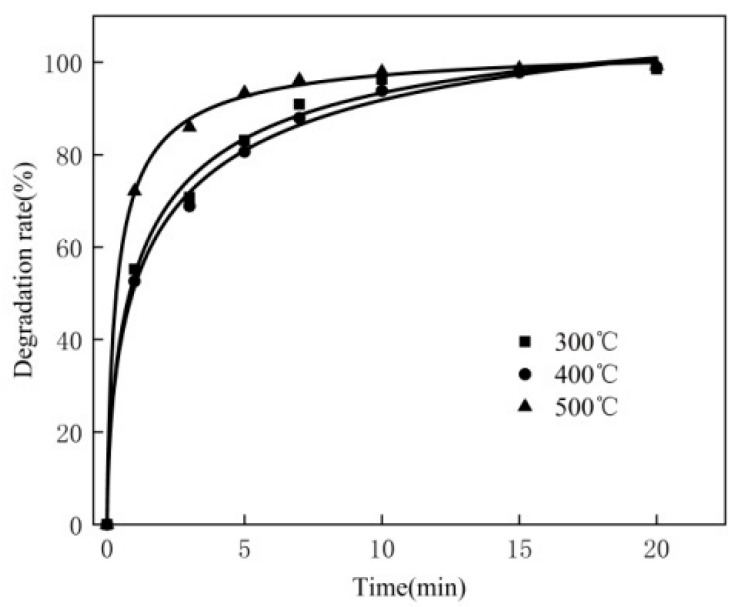
Effect of calcination temperature.

**Figure 7 materials-16-02087-f007:**
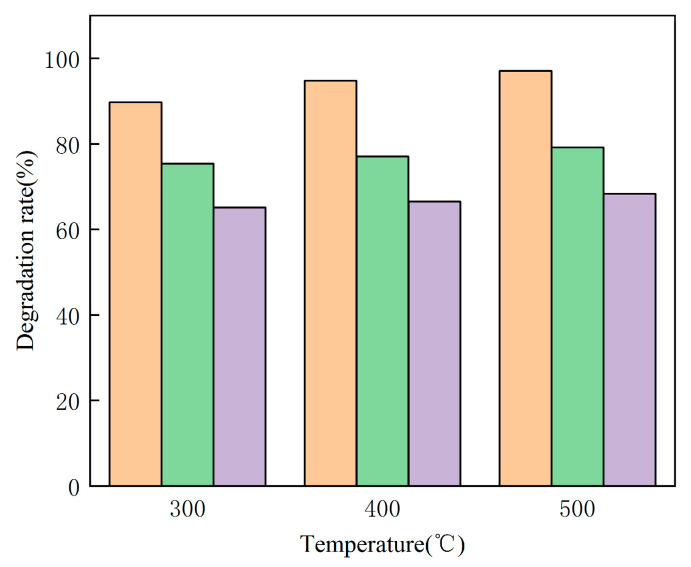
Regeneration experiment.

**Figure 8 materials-16-02087-f008:**
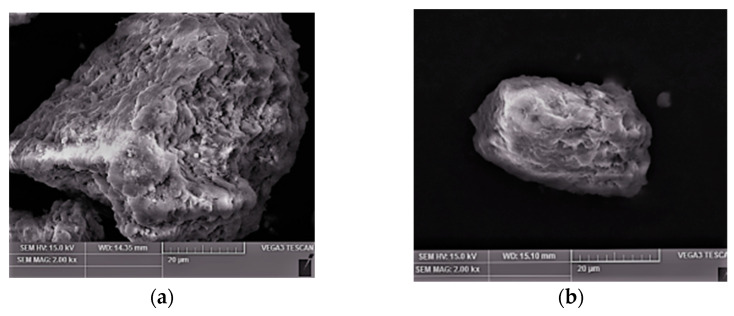
(**a**) SEM image of original soil, (**b**) modified catalyst, (**c**) original soil, (**d**) modified catalyst.

**Figure 9 materials-16-02087-f009:**
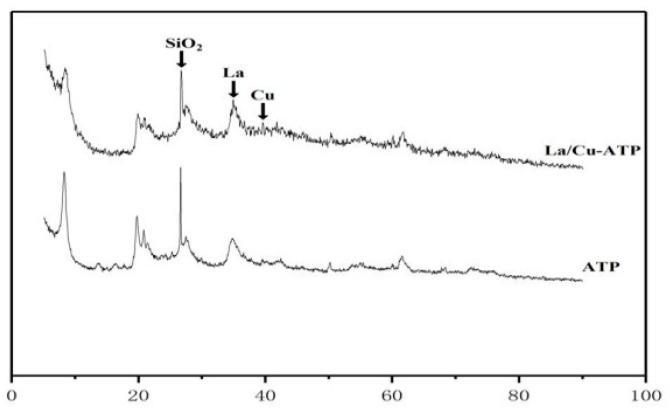
XRD of La^3+^/Cu^2+^-ATP.

**Figure 10 materials-16-02087-f010:**
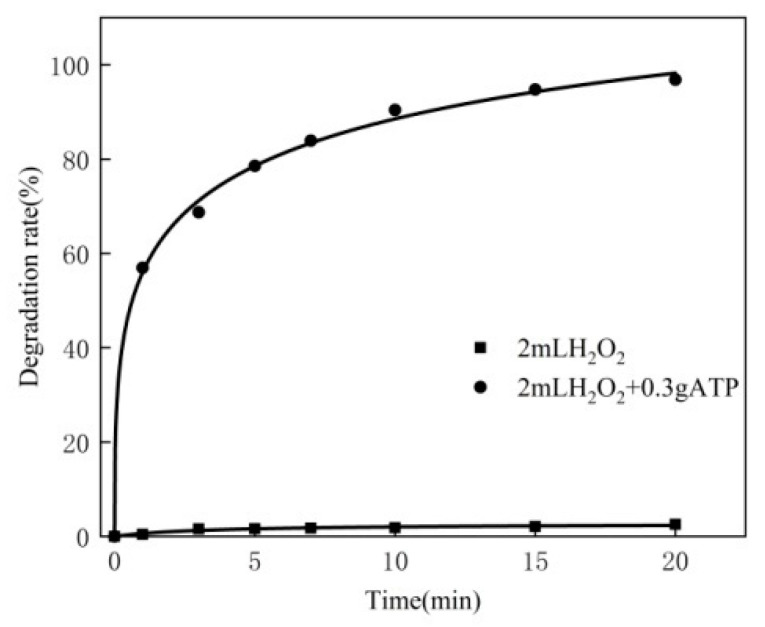
Experimental results of H_2_O_2_.

**Figure 11 materials-16-02087-f011:**
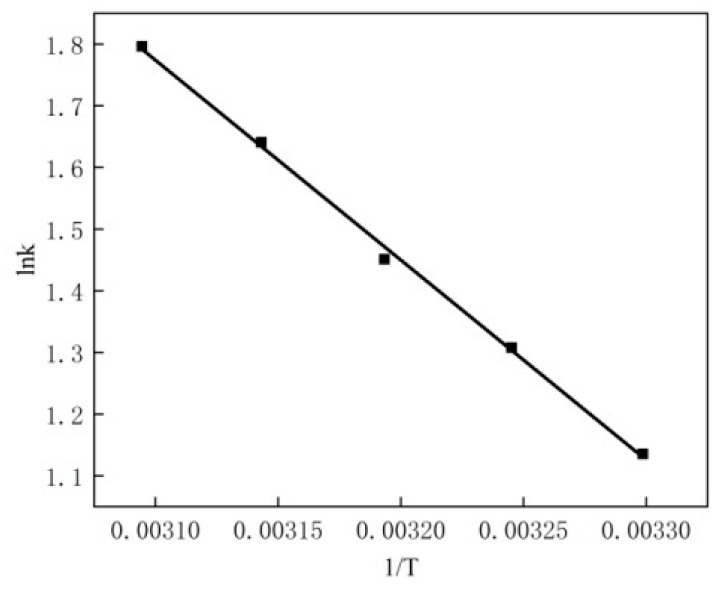
lnk-T^−1^.

**Table 1 materials-16-02087-t001:** k values at different reaction temperatures.

Temperature/°C	k
30	3.112
35	3.698
40	4.267
45	5.159
50	6.027

## Data Availability

The authors confirm that the data supporting the findings of this study are available within the article.

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
