# Peer review of "Kinetics of Catalytic Oxidation of Methylene Blue with La/Cu Co-Doped in Attapulgite"

_materials, 2023, doi:10.3390/ma16052087_

Round 1

Reviewer 1 Report

This manuscript describes the catalytic oxidation of methylene blue wastewater with attapulgite dopped with La3+ and Cu2+. The work has been carried out with care and the results have been presented with clarity and discussed appropriately. In addition, the catalyst was reused several times obtaining 80 % of degradation of methylene blue. In this way, even though this manuscript demonstrates a significant contribution in the area of materials, it is necessary solving several issues. Accordingly, I can recommend publication of this work in Materials with major revisions.

Issues:

- The authors should check formatting errors in the manuscript.

- This investigation/manuscript should make a comparison of its results with the bibliography and demonstrate the significant improvement and advantage of its results.

- This investigation/manuscript should make a study of more cycles of reuse of the catalyst and improve it.

- The authors should rewrite the conclusions without using points.

Author Response

Thank you very much for your correct review comments. I have made detailed and serious modifications to each question, and the contents are as follows:

Comment No. 1:  The authors should check formatting errors in the manuscript.

Response: It has been carefully revised according to the format requirements of magazine articles, especially the format of references.

Comment No. 2: This investigation/manuscript should make a comparison of its results with the bibliography and demonstrate the significant improvement and advantage of its results.

Response: A document has been added to compare the experimental results. The added documents is as follow:

[30]Yin L, Lu X, Ai F. Effect of Ti-Attapulgite Catalyst on Ozonation Degradation of Dye Wastewater. J. Chin. Ceram. Soc. 2003, 31 (1): pp. 66–69.

Comment No. 3:This investigation/manuscript should make a study of more cycles of reuse of the catalyst and improve it.

Response: The datas of MB degradation rate after third use of catalyst has been added, and the experimental results were analyzed.

Comment No. 4: The authors should rewrite the conclusions without using points.

Response: The conclusion part has been reorganized and the redundant part has been deleted.

Reviewer 2 Report

General Comments.
1. The document, when analyzed on Plagiarisms software, i.e., Turnitin, is showing 17%. As per my view, it must be lower down up to 14% or so for an article. Self-Plagiarisms are also not accepted more than 2.5%.
2. The tables and figures used are not clear and can be enhanced. Heading must be with sequential numbers like 1.0, 1.1, 1.2, etc.
3. In reference section some reference is de-shaped, may be due to formatting. They are also needed to be corrected as per journal format.
4. The introduction must be reduced to one and a half pages.
5. The title needed significant modification.
6. The numbering of content must correct.
7. The manuscript requires an extension of the literature.
8. The manuscript does not illustrate great attention and activity in the field.
9. Tables also contain few references.
10. Please enhance the manuscript on analysis of earlier mention issues.
11. The figure number is distorted and can be rechecked.
12. For the text clarity, would you refrain from using additional words, mostly meaningless filler words, which can be omitted or some archaic words see, e.g. "respectively", "thus", "hence", "therefore", "furthermore", "thereby", "basically,", "meanwhile", "wherein", "herein", "Nonetheless", "Perceivably," etc.?

Specific Comments.

Abstract: This section should be reorganized for consistency. It could also be shortened by deleting and synthesizing 2 or 3 sentences, or summarizing certain ideas.
- I suggest to present the original contributions after synthesizing the main results
- Please, specify the methodology after the research questions.
- The aim of the study could come after the literature gap and before the theory

Introduction: This section should be improved and supported by more and recent references. The authors cannot ignore the precursors of (neo-)institutional theory and work in this field. The authors must clearly demonstrate the relevance of the subject and the problem, justify the literature gap, show the main results and the originality of the contribution. The authors should describe the importance of their research more clearly. The references cited lack articles on emerging contaminants from last year. So, add more references (2014-2021) to support the author's points of view. Last paragraph must be an outline of the complete study showing the needed and targets assumed in the paper. Hence need minor revision. It also suggestive to add latest article in references. Please use the literature background on sustainability (but not self-citations, please) to broaden the manuscript foundation. Please develop a better title. This one does not state what is important to catch the prospective reader's attention. Some important references relevant to the study which can be added.

Abstract: It is needed to be started with small introduction and then quantitative description of the paper. It is also suggestive to shorten few unnecessary sentences in abstract. Underscore the scientific value-added to your paper in your abstract. Please look at articles we have published for models. Your abstract should clearly state the essence of the problem you are addressing, what you did and what you found and recommend. That would help a prospective reader of the abstract to decide if they wish to read the entire article.
2.1 Materials and Reagents.: More specific details are needed to be added with use of latest reference. Better use recent article for updating this section. Figure 1,2 and Fig. 4 need more elaboration in text. To be convincing, the methodology should be justified and more detailed.

3. Results and discussion: Discussion and conclusions clearly do not establish a strong correlation with environmental concerns (however, as much as possible, avoid self-citations). In your discussion section, please link your empirical results with a broader and deeper literature review. Discussions and conclusions must go deeper, it would be more interesting if the authors focus more on the significance of their findings regarding the importance of the interrelationship between the obtained results and sustainable development  in the sector context, and the barriers to do it, what would be the consequences, in the real world, in changing the observed situation, what would be the ways, in the real world, to change/improve the observed situation.

Future scope of this study can be added as well as social impact can also be discussed in this paper.

Conclusions: This section is needed to be free from any variables are symbols. Only main pointed like what was expected and what was achieved must be written. What signification contribution this study to the society must be mentioned in this section. Please make sure your conclusions' section underscores the scientific value-added of your paper and/or the applicability of your results. Highlight the novelty of your study. Clearly discuss what the previous studies that you are referring to are. What are the Research Gaps/Contributions? In your conclusions, please discuss the implications of your research.

Author Response

Dear Reviewer:

Thank you very much for your correct review comments. I have made detailed and serious modifications to each question, and the contents are as follows:

Comment No. 1: The document, when analyzed on Plagiarisms software, i.e., Turnitin, is showing 17%. As per my view, it must be lower down up to 14% or so for an article. Self-Plagiarisms are also not accepted more than 2.5%.

Response: Our research group has been committed to the research in this field and has published a series of articles. There will be some repetitive parts in the experimental method, but the data and result discussion are completely different. I have rewritten the experimental part to maintain a relatively low repetition rate.

Comment No. 2: The tables and figures used are not clear and can be enhanced. Heading must be with sequential numbers like 1.0, 1.1, 1.2, etc.

Response: According to the requirements of the journal, the figures and tables are numbered separately, and the numbers of the figures in the full text have been sorted according to the requirements.

Comment No. 3:  In reference section some reference is de-shaped, may be due to formatting. They are also needed to be corrected as per journal format.

Response: It has been carefully revised according to the format requirements of journal references.

Comment No. 4:  The introduction must be reduced to one and a half pages.

Response: The introduction has been reorganized, and the redundant part has been deleted.

Comment No. 5: The title needed significant modification.

Response: The title has been modified to "Kinetics of catalytic oxidation of methylene blue with La/Cu co-doped in attapulgite"

Comment No. 6: The numbering of content must correct.

Response: The numbering of content has been confirmed again.

Comment No. 7: The manuscript requires an extension of the literature.

Response: Three documents have been added for supplement.

  1. Muhammad I, Ali H, Syeda T. B. et al. Synthesis of Al/starch co-doped in CaO nanoparticles for enhanced catalytic and antimicrobial activities: experimental and DFT approaches. RSC Adv. 2022; 12: pp. 32142–32155.
  2. Sawaira M, Muhammad I, Ali H. et al. Comparative Study of Sonophotocatalytic, Photocatalytic, and Catalytic Activities of Magnesium and Chitosan-Doped Tin Oxide Quantum Dots. ACS Omega 2022; 7: pp. 46428−46439.
  3. Yin L, Lu X, Ai F. Effect of Ti-Attapulgite Catalyst on Ozonation Degradation of Dye Wastewater. J. Chin. Ceram. Soc. 2003, 31 (1): pp. 66–69.

Comment No. 8: The manuscript does not illustrate great attention and activity in the field.

Response: I have added the sentences to the introduction “ATP is suitable for catalyst carrier for its large specific surface area, high chemical stability and mechanical stability. Especially for the low price, ATP has practical significance in wastewater treatment.”

Comment No. 9: Tables also contain few references.

Response: This issue has been confirmed again.

Comment No. 10: Please enhance the manuscript on analysis of earlier mention issues.

Response: Some analyses have been supplemented and modified.
Comment No. 11: The figure number is distorted and can be rechecked.

Response: The wrong figure number has been modified.

Comment No. 12: For the text clarity, would you refrain from using additional words, mostly meaningless filler words, which can be omitted or some archaic words see, e.g. "respectively", "thus", "hence", "therefore", "furthermore", "thereby", "basically,", "meanwhile", "wherein", "herein", "Nonetheless", "Perceivably," etc.?
Response: The additional words have been deleted.

Reviewer 3 Report

This manuscript reports a study on “Catalytic oxidation of methylene blue wastewater with atta- 2 pulgite/(La3++Cu2+)”. Some major issues need to be addressed before the manuscript can be considered for publication

1.     The authors should move the last part of the introduction section to the conclusions part and also support it with recommendations and future perspectives.

2.     Please provide a detailed analysis of XRD spectra (including line width/position/areas ratios analysis).

3.     The organization of paper needs to be improved.

4.     The authors are requested to add graphical abstract to support the manuscript.

5.     Mineralization degree of MB dye before and after the reaction is important to investigate. The authors should add this experiment to the text.

6.     Why authors used MB dye as degradation. Give any specific reason

7.     Research article still lacking with recent literature authors should read below recent relevant reference and consider for citation

https://doi.org/10.1039/D2RA06340A and https://doi.org/10.1021/acsomega.2c05133.

Author Response

Dear Reviewer:

Thank you very much for your correct review comments. I have made detailed and serious modifications to each question, and the contents are as follows:

Comment No. 1:     The authors should move the last part of the introduction section to the conclusions part and also support it with recommendations and future perspectives.

Response: the last part of introduction section has been moved to the conclusions part, and modified carefully.

Comment No. 2: Please provide a detailed analysis of XRD spectra (including line width/position/areas ratios analysis).

Response: The detail of XRD spectra has been analyzed.

Comment No. 3: The organization of paper needs to be improved.

Response: The content has been adjusted appropriately, for example, the part of the preface has been moved to the conclusion part

Comment No. 4:  The authors are requested to add graphical abstract to support the manuscript.

Response: We have added a graphical abstract as follows:

Fig. Graphical abstract

Comment No. 5: Mineralization degree of MB dye before and after the reaction is important to investigate. The authors should add this experiment to the text.

Response: The purpose of this paper is to obtain the kinetic model of MB degradation, mainly through investigating the effects of some factors, such as reaction temperature, pH and methylene blue (MB) concentration on the reaction rate of MB. The results can provide theoretical support for the design and industrial application of subsequent reactors. The study of mineralization degree of MB will be an important content of the next work.

Comment No. 6: Why authors used MB dye as degradation. Give any specific reason

Response: Methylene blue (MB) is a common pollutant in wastewater of printing and dyeing industry. So far, few literatures have established degradation kinetic models based on the degradation mechanism of MB. The establishment of its kinetic model is conducive to the industrial application of the technology.

Comment No. 7: Research article still lacking with recent literature authors should read below recent relevant reference and consider for citation

https://doi.org/10.1039/D2RA06340A and https://doi.org/10.1021/acsomega.2c05133.

Response: Two documents have been added as follows:

  1. Muhammad I, Ali H, Syeda T. B. et al. Synthesis of Al/starch co-doped in CaO nanoparticles for enhanced catalytic and antimicrobial activities: experimental and DFT approaches. RSC Adv. 2022; 12: pp. 32142–32155.
  2. Sawaira M, Muhammad I, Ali H. et al. Comparative Study of Sonophotocatalytic, Photocatalytic, and Catalytic Activities of Magnesium and Chitosan-Doped Tin Oxide Quantum Dots. ACS Omega 2022; 7: pp. 46428−46439.

Round 2

Reviewer 1 Report

This manuscript describes the catalytic oxidation of methylene blue wastewater with attapulgite dopped with La3+ and Cu2+. The work has been carried out with care and the results have been presented with clarity and discussed appropriately. In this way, this manuscript demonstrates a significant contribution in the area of materials. Accordingly, I can recommend publication of this work in Materials. The authors have correctly made the changes suggested by the reviewers.

Reviewer 2 Report

No further comments